# Tracking Migraine Symptoms: A Longitudinal Comparison of Smartphone-Based Headache Diaries and Clinical Interviews

**DOI:** 10.3390/neurolint17030033

**Published:** 2025-02-24

**Authors:** Nicolas Vandenbussche, Jonas Van Der Donckt, Mathias De Brouwer, Bram Steenwinckel, Marija Stojchevska, Femke Ongenae, Sofie Van Hoecke, Koen Paemeleire

**Affiliations:** 1Department of Neurology, Ghent University Hospital, Corneel Heymanslaan 10, B-9000 Ghent, Belgium; 2Department of Basic and Applied Medical Sciences, Faculty of Medicine and Health Sciences, Ghent University, Corneel Heymanslaan 10, B-9000 Ghent, Belgium; 3Department of Neurology, AZ Sint-Jan Brugge, Ruddershove 10, B-8000 Bruges, Belgium; 4IDLab, Ghent University—imec, Technologiepark-Zwijnaarde 126, B-9052 Ghent, Belgium; jonvdrdo.vanderdonckt@ugent.be (J.V.D.D.); bram.steenwinckel@ugent.be (B.S.); femke.ongenae@ugent.be (F.O.); sofie.vanhoecke@ugent.be (S.V.H.)

**Keywords:** migraine, smartphone, mHealth, headache, diary, prospective

## Abstract

**Background/Objectives:** By leveraging the capabilities of a smartphone-based headache diary, the objective of this study was to determine the amount of agreement between migraine-associated symptomatology during headache events and the symptoms documented during clinician-led intake interviews. **Methods**: This was a 90-day longitudinal, smartphone-based headache calendar study for participants diagnosed with migraine. Registered headache events were labeled as “definite migraine”, “probable migraine”, and “not migraine” in accordance with the International Classification of Headache Disorders, Third Edition (ICHD-3) criteria. Symptoms’ agreement with clinician-led intake interviews (agreement percentages and kappa coefficients), symptoms’ similarity between headache events within users (percentage), and amount of newly registered ICHD-3 symptoms per participant were calculated. **Results**: Twenty-seven participants provided 505 headache events eligible for analysis. The median agreement between recorded headache event symptomatology and clinician-led intake interview phenotyping ranged between 40% (for events fulfilling “not migraine” criteria) and 55.5% (“definite migraine”) (*p* < 0.001). Higher intraparticipant headache event pair similarity was observed for “definite migraine” pairs (*p* < 0.01), along with a decreasing trend in similarity as the attack-pair headache distance increases. Over half of the participants registered at least one new ICHD-3 symptom during the study. **Conclusions**: Electronic diary registrations show substantial longitudinal variability in intrapersonal headache symptomatology, with the similarity of headache events declining over time. The registration of a new ICHD-3 symptom was the rule rather than the exception.

## 1. Introduction

Migraine is a common neurological disorder characterized by recurring headache events, lasting several hours to three days and accompanied by symptoms, such as photophobia, phonophobia, nausea, or vomiting [1,2]. The symptomatology of migraine is not restricted to symptoms listed in the International Classification of Headache Disorders (ICHD), but expands widely over many associated sensory, cognitive, homeostatic, and autonomous symptoms [3]. The disorder affects more than one billion people worldwide and brings a high level of burden to patients, many of whom suffer from reductions in quality of life, and predominantly affects individuals during their most productive years [4]. Migraine is a neurological disorder involving the trigeminovascular system and various central networks of the brain. The pathophysiology involves disturbances in sensory processing, wherein lack of habituation to stimuli and sensitization of various neural structures play key roles [2,5].

Diagnosing and treating migraine presents inherent challenges, primarily due to its symptomatology exhibiting substantial interpatient and intrapatient variability [6]. In headache medicine, history taking is a fundamental part of patient care. Headache event-related characteristics and symptomatology must be conveyed from patient to healthcare provider through various means of communication. The information transfer during patient visits, therefore, is subjective and may be only partial or incomplete due to time restrictions, recall bias, and/or questions not being asked by the physician [7,8]. Headache diaries or calendars have been the mainstay of clinical follow-up for patients with migraine [8]. These tools provide valuable data on days with headache or migraine, symptomatology, and trigger factors (e.g., the menstrual cycle) and add to the clinical information space by combining longitudinal information with the standard history taking [7]. Paper diaries have been used by patients for many years to record individual headache event characteristics, but this approach poses additional limitations, including potential information loss, inconsistency in entries, and cumbersome data retrieval [9].

With recent technological advancements, patients now have the option to utilize dedicated smartphone headache diary applications within the context of ecological momentary assessment (EMA) [10,11,12,13,14]. EMA is the scientific term for the repeated sampling and registering of a subject’s behaviors and experiences of, e.g., a chronic disorder in real time or near real time in the subject’s natural environments [10]. As such, EMA can provide more objective data in addition to subjective self-reporting during classical direct patient–physician interactions. EMA may provide a means for diagnostics, further exploration of the nosology of disorders, and follow-up of the burden of disorders. Digital tools for EMA offer the potential for more efficient, structured, and comprehensive self-monitoring of the disorder, enabling patients to collect context information on various aspects of the condition and its treatment, while facilitating the sharing of pertinent information with healthcare providers [15]. Electronic diaries are being implemented widely in medicine [16,17,18]. In headache medicine, it is worth noting that certain studies have indicated a patient preference for electronic headache diaries over traditional paper-based methods [19].

Previous studies have looked at the characteristics of singular headache events in individual patients and the evolution of headache events’ symptomatology through time [6,20,21]. These analyses are subject to undertaking a comprehensive investigation of several facets, including the comparison of symptomatology across distinct headache events, the alignment of headache event-related symptomatology with data collected during clinician-led headache intake interviews, and the longitudinal tracking of newly discovered symptoms by the participants [6,20,21]. The largest study to date by Verhagen et al. compared differences in migraine attack characteristics between men and women longitudinally through electronic headache diaries and found that compared with attacks in men, both perimenstrual and non-perimenstrual migraine attacks are of longer duration and are more often accompanied by associated symptoms [22]. This study shows the capability of electronic headache diary studies to find latent aspects of migraine symptomatology. However, there still remains a research gap in the heterogeneity and variability of intraparticipant extended migraine symptomatology over time from longitudinal studies that apply smartphone-based headache diaries.

Leveraging the capabilities of a smartphone-based headache diary, our study analyzed the heterogeneity and variability of intraparticipant migraine symptomatology over an extended period of 90 days. The primary objective of this study was to determine the amount of agreement between migraine-associated symptomatology during headache events and the symptoms documented during clinician-led intake interviews. The secondary objectives were (i) the amount of intraparticipant similarity in symptomatology between headache events, with a specific focus on the temporal distance between both events, and (ii) the proportions of newly experienced ICHD Third Edition (ICHD-3) symptoms by users over time and whether certain ICHD-3 symptoms tend to be over- or underreported. We hypothesized moderate intake symptom agreement and low to moderate intraparticipant event similarity, with a decline in intraparticipant similarity as the time between events increases.

## 2. Materials and Methods

This was a 90-day observational, longitudinal, smartphone-based headache calendar study for participants diagnosed with migraine. The analysis is part of the “mBrain” study, a comprehensive and longitudinal research initiative as detailed in De Brouwer et al. [12]. The goal of the “mBrain” study is to provide a profound understanding of the manifestations of migraine in patients within ambulatory environments. To accomplish this, participants in the study were equipped with wrist-worn wearable devices, specifically the Empatica E4^®^ (Empatica Srl, Milano, Italy), to capture relevant physiological data (e.g., galvanic skin response, heart rate) for at least 90 days. The wearable was connected via Bluetooth to a dedicated headache diary smartphone application developed by academic experts. Within this mobile application, participants could indicate their headache periods, headache event-associated characteristics (e.g., location of pain, intensity, symptomatology, etc.), medication intakes, and behavior in terms of food intake. Their physical activities, sleep, and stress-related events were automatically captured by using machine learning on the wearable sensor data [12]. This unique setup facilitated continuous ambulatory data collection and offered a deeper insight into the various facets of migraine and its symptomatology over an extended period. This study focused on analyzing headache events’ symptomatology recorded via the smartphone application and did not include the analysis of wearable-derived physiological data.

### 2.1. Participants

Participants were included in the study if they were between 18 and 65 years of age with a history of migraine longer than one year. The diagnosis of migraine was made by neurologists with expertise in the field of headache disorders. Participants with either episodic or chronic migraine according to the ICHD-3 criteria could partake in the study. Other inclusion criteria were, on average, a minimum of 5 crystal-clear days per month without headache or headache-related symptoms, and onset of migraine before the age of 50. Participants also had to use an Android-based smartphone (Android Operating System (OS) version 8.0 or greater). The exclusion criteria were any other diagnosed headache disorder apart from tension-type headache, the presence of medication overuse headache (ICHD-3 8.2 and subsections), daily persistent headaches or headache-related symptoms, the presence of any other chronic pain syndrome, significant medical comorbidity deemed by the physician researcher to interfere with the study, the presence of opioid or barbiturate usage, the presence of illicit drug or alcohol abuse, and the presence of pregnancy or immediate pregnancy plans. Participants could not simultaneously participate in any other academic or commercial medical study. Participants were allowed to use acute or preventive treatment as required from a clinical perspective, with no plan to switch preventive treatment during the course of the study.

### 2.2. Study Design

All study participants were interviewed during the intake visit of the study by a physician researcher and neurologist with expertise in the field of clinical headache disorders (N.V.). Together with the physician researcher, the participants went through a list of possible headache-associated symptoms and answered whether the symptom was present before, during, and/or after typical migraine headache events or not (multiple answers possible). This intake symptomatology list was represented by a questionnaire within the “mBrain” smartphone application (Figure 1).

During the study, participants used the “mBrain” application as an electronic headache diary, which also prompted daily morning and evening questionnaires to verify the correctness of the reported headache and medicine intake events of the past day. The app also assessed regular health updates and/or predictions of the machine learning models via ad hoc questionnaires, such as subjective sleep quality, stress level, and mood. Whenever a participant experienced a headache event, the participant entered the symptoms and headache event’s characteristics in the headache event registration module of the mBrain application. Apart from the associated symptoms, the participants also recorded the start and end time of the headache event (format: year–month–day hour:minutes), the location of the pain, the intensity of the pain (no pain, mild pain, moderate pain, severe pain, very severe pain), triggers, associated symptoms (without specifying the particular phase during which the symptom occurred), medication taken, and whether medication usage was successful (Figure 2). During the participation of subjects in the study, the investigators had access to data on compliance with the smartphone application (to determine whether there were any problems or to encourage participants to continue with the trial), but did not have access to any clinically relevant data.

This symptomatology list is divided into ICHD-3 symptoms and non-ICHD-3 symptoms. For the list of ICHD-3 symptoms, we used the wording and definitions within the ICHD-3 classification of ICHD-3 1.1 “migraine without aura” and ICHD-3 3.1 “cluster headache”. We also used the definitions within the ICHD-3 list of definitions of terms. As such, the ICHD-3 symptom list consisted of the following symptoms: conjunctival injection, eyelid edema, forehead and facial sweating, lacrimation, miosis, motion sensitivity, nasal congestion, nausea, pain aggravation during routine physical activity, phonophobia, photophobia, ptosis, restlessness or agitation, rhinorrhea, throbbing headache, and vomiting.

The list of non-ICHD-3 symptoms was carefully constructed on the basis of medical literature related to extended migraine symptomatology retrieved from extensive migraine phenotyping [19,23,24,25,26]. It consisted of the following symptoms: anxiety, constipation, craving sweet or salty food, decreased ability to make sentences, decreased ability to remember, decreased ability to speak, decreased interest in daily activities, depressed mood, diarrhea, excessive thirst, feeling exhausted, fatigue, feeling elated or happy, flushed face, frequent urination, hoarseness, hypersalivation, hypersensitivity of face skin, hypersensitivity of neck skin, hypersensitivity of scalp skin, impaired concentration, irritability, lightheadedness, loss of appetite, oliguria, osmophobia, pale face, polyuria, stiffness of neck, stomach ache, swollen feeling in mouth, swollen feeling in throat, urge to move, internal vertigo (“spinning sensation as if your body is falsely moving”), external vertigo (“spinning sensation as if the surroundings are moving”), and yawning.

The mobile application adopted a unique approach for inputting and processing data related to the ICHD-3 symptom list, which differed from the method used for non-ICHD-3 symptoms. Participants were always able to select from the above-listed ICHD-3 symptoms when recording a headache episode in the longitudinal phase. In contrast, participants were only able to select a specific non-ICHD-3 symptom if they had indicated during the clinician-led intake interview that they might experience this non-ICHD-3 symptom before, during, or after a typical migraine headache event. This intake-generated headache event symptomatology list approach was chosen a priori (i.e., during study design development) to improve headache symptom registration user experience (i.e., to not overwhelm the participants with a very long list of symptoms, difficult to process on the smartphone). Additionally, participants had the option to document new symptoms, which could include those not listed among the non-ICHD-3 symptoms, in a free text entry format. Detailed summary statistics for these manually entered symptoms are provided in Appendix A.

The ICHD-3 symptom “Pain aggravation during routine physical activity” was not asked within the intake questionnaire due to unintended omission but was available during headache event registration. For the “throbbing headache” symptom, the intake data of one participant are not present.

The study design did not preset a fixed sample size, nor did it calculate a sample size through power calculation. The mBrain study was an observational pilot study, where the aim was to include as many participants with migraine as possible. As this study only investigated events related to migraine attacks, a control group of participants without migraine was not recruited.

### 2.3. Data Processing 

Headache event symptomatology was registered during the clinician-led intake interview and the longitudinal registration phase as follows. During the clinician-led intake interview, a symptom was determined as “present” if it was reported as occurring in at least one phase (the premonitory, ictal, or postdromal phase) of the migraine headache event. During the longitudinal phase, each available symptom could be assessed individually during headache event registration, categorizing them as either “present” or “not present” for each individual headache event.

Single registered headache events were checked according to Criteria B (excluding the requirement of untreated or unsuccessfully treated headache events), C, and D for fulfillment of the ICHD-3 criteria for migraine without aura (ICHD-3 1.1). When an event satisfied Criteria B, C, and D, the group label “definite migraine” was given. If not, the headache event was checked for the fulfillment of Criterion A for probable migraine (ICHD-3 1.5), and, if fulfilled, the group label “probable migraine” was given. When the headache event did not fulfill the label “definite migraine” or “probable migraine”, the group label “not migraine” was given.

To ensure accurate analysis of recorded headache symptoms, the entry time metadata of the headache event records were utilized to identify any potential reporting bias. First, we excluded headache events registered more than 24 h after the headache’s reported end time due to a high possibility of recall bias affecting the accuracy of the event’s symptomatology. Second, we identified a risk for predictive bias when the record was made before the headache was ended, potentially leading to the omission of later-occurring symptoms. To address this, we excluded records entered either more than two hours before the headache’s end or earlier than 25% of the total headache duration, whichever was longer. Headache records that did not adhere to both these criteria were excluded from all subsequent analyses.

### 2.4. Statistical Analysis and Outcome Parameters

Baseline characteristics of the participants (age, sex, average migraine days per month, years lived with migraine, days in study) are presented as the means and standard deviations (SD) for continuous variables, and raw counts and percentages for categorical variables. Compliance rate with the daily questionnaire, which functions as a proxy for the amount of daily interaction with the application, was defined as the percentage of days an individual participant completed the daily morning or evening questionnaire, with the median and interquartile range (IQR) (first (Q1) and third quartiles (Q3)) for all participants presented.

Symptom agreement was defined as consistency between the symptom’s presence or absence during the clinician-led intake interview and its presence or absence during single headache event registration, denoted as “present” (intake) and “present” (headache event), or “not present” (intake) and “not present” (headache event). In all other cases, it was considered a disagreement. To assess agreement differences between group labels, we applied the Mann–Whitney U test for each group label combination (e.g., “definite migraine” vs. “not migraine”). For the ICHD-3 symptoms list, we measured Cohen’s kappa coefficient of repeatability [27]. As documented above, all ICHD-3 symptoms could be consistently registered by all participants during the longitudinal phase, whereas participants had different subsets (as a result of the personalization) of the full non-ICHD-3 symptoms list available during headache event registration. This limits the agreement analyses of the non-ICHD-3 symptoms list solely to the symptoms determined by “present” (intake) and “present” (registration), as the “not present” (intake) symptoms are not available within the participant’s non-ICHD-3 symptom list. Non-ICHD-3 symptoms that were not available within the intake-generated headache event registration list were thus excluded for agreement analysis for the given participant.

The similarity between intraparticipant headache events is determined using the Intersection over Union (IoU) metric, which compares the reported symptoms of two headache events with the same group label. The IoU similarity is preferred over the typical percentage pair-agreement because it minimizes the impact of symptoms not reported in the headache pair. Percentage pair-agreement considers both the presence and absence of symptoms, which can be skewed if the participants report very few symptoms. For instance, a patient’s intake-generated symptom list includes 30 symptoms, and suppose that the patient registers 2 subsequent headache events with each 3 symptoms, then the maximum symptom mismatch between these two headache events is six. This results in a minimum similarity of 80% for the percentage pair-agreement (as the 24 other symptoms stay absent 24/30 = 80%), whereas the IoU similarity will only consider the reported symptoms.

In addition, we aimed to analyze how the similarity of headache events varies over time, focusing on the intervals between headache event pairs. To do so, we introduced the concept of a “headache interval distance”, denoting the time delta between pairs of headache events. Given N total eligible headache events, there are N × (N − 1)/2 headache event pairs, thus scaling quadratically with N. As such, participants with more headache events of a label group have quadratically more pairs compared with those with fewer headache events.

To account for this user imbalance regarding the amount of headache pairs, and also ensure equal weighting of pairs over time, we aggregate headache pairs for each participant in 5-day bins based on the “headache interval distance” and use the median IoU similarity values per bin instead. Analyzing the available pairs for these bins, illustrated by Figure 3, demonstrates a right-skewed sample availability, with very few samples for the “definite migraine” and “probable migraine” groups for large headache interval distances. As such, we opted to limit our analysis to samples with a headache interval distance smaller than 60 days, using non-overlapping bins of 5 days, resulting in a total of 12 bins. These median aggregated bin values are represented per participant and ICHD-3 attack scoring group, with a regression line and a 95% confidence interval (CI) fitted per attack scoring group. This methodology is repeated for the ICHD-3 and non-ICHD-3 symptom subset of each headache event–label pair.

We present the median of the per-participant mean percentage IoU similarity, with first quartile (Q1) and third quartile (Q3) numbers, for each ICHD-3 scoring group.

Lastly, newly registered ICHD-3 symptoms, i.e., symptoms not reported during intake, are presented numerically and with the percentage of participants.

All data analysis and statistical tests were performed in Python version 3.8, utilizing the SciPy package for statistical testing and the seaborn toolkit for data visualization [28,29].

### 2.5. Ethical Approval and Patients’ Consent

The study was approved by the Ethics Committee of University Hospital Ghent (BC-10031). The study was preregistered at clinicaltrials.gov (NCT04983186). All patients gave informed consent for the collection and analysis of their pseudonymized data, and for publication of the results.

## 3. Results

### 3.1. Baseline Characteristics

Thirty (*n* = 30) participants were included in the study. Two participants could not start the study due to technical issues between the app and the participant’s smartphone. One participant did not provide a minimum of one headache event during the duration of the study. In total, 27 participants provided results for this study.

The average age of participants (*n* = 27) was 37 years (standard deviation (SD) 13). Twenty-one participants (78%) were of the female sex. Participants had a mean of 7.7 migraine days per month (SD 5.5) and had a mean length of headache history of 18 years (SD 11) (Table 1).

A total of 572 headache events were recorded. The median number of registered headache attacks per participant was 15 (Q1–Q3 5.5–34). After excluding events with potential predictive or recall bias (as described in the methodology section), 505 eligible individual headache events (from 27 participants) remained for analysis, with a median number of registered headache attacks per participant of 15 (Q1–Q3 4.5 to 30.5). Of these, the median of the per-participant average acute medication use was 87.5% (Q1–Q3 52.5–100%), with a therapeutic effectiveness of 80% (Q1–Q3 50.0–100%). For these 505 headaches, 74 events (14.9%) were classified as “definite migraine”, 155 headache events (30.7%) as “probable migraine”, and 276 headache events (54.7%) as “not migraine” (Table 2). The “definite migraine” group label had a higher number of both ICHD-3 and non-ICHD-3 symptoms compared with “probable migraine”, while the “not migraine” group label had overall the lowest number of reported symptoms (Table 2).

Appendix A presents the frequency of specific symptoms, including both ICHD-3 and non-ICHD-3 symptoms, experienced by participants at various stages of individual migraine events, as reported during intake. Additionally, Appendix A displays the average agreement for each symptom.

### 3.2. Agreement of Headache Event Symptomatology with Clinician-Led Intake Interviews

For all headache events fulfilling the criteria for the group label “definite migraine”, the median symptomatology agreement with clinician-led intake interviews was 55.5% (Q1–Q3 38.9–71.6%, *n* = 74) (using all available symptoms), as shown in Figure 4A. For “probable migraine” events, this median symptomatology agreement was 41.9% (Q1–Q3 34.9–55.5%, *n* = 155) (Figure 4B), while for those labeled as “not migraine” events, this agreement was 39.4% (Q1–Q3 32.3–51.7%, *n* = 276), as shown in Figure 4C.

A statistically significant difference between events with group label “definite migraine” and the two other groups was found (Mann–Whitney U test, *p* < 0.01 for both tests for all symptom subsets (Figure 4A–C).

Disagreement between headache event symptomatology and the information from clinician-led intake interviews was mostly driven by non-ICHD-3 symptoms (Figure 4 and Table 3). Note that the non-ICHD-3 symptom agreement is solely determined by “present” (intake) and “present” (registration), as explained in the methodology. As such, each headache event’s non-ICHD-3 agreement is directly determined by the ratio of reported non-ICHD-3 symptoms.

The agreement trend analysis does not indicate a trend change, considering the 95% confidence interval (Figure 4, subplot D–F). Similarly, there is no discernable trend in the overall number of reported symptoms (Figure 4, subplot G–I), except that “definite migraine” and “probable migraine” attacks have more reported symptoms than the “not migraine” group, which can also be noted in Table 2. Additionally, there is a notable increase in reporting both ICHD-3 and non-ICHD-3 symptoms at the beginning and end of the study period for the “definite migraine” and “probable migraine” label groups.

Subplot J presents the number of (eligible) registered events for each group label, aggregated into 5-day intervals relative to the study onset. A decreasing trend is observed for the “not migraine” label group as the study progresses, whereas this trend is less apparent for the “probable migraine” and “definite migraine” label groups.

For the list of ICHD-3 symptoms, the mean Cohen’s kappa coefficient of repeatability between clinician-led intake interviews and headache events was 0.48 (SD 0.27) for headache events fulfilling the criteria for “definite migraine” (moderate agreement), 0.32 (SD 0.24) for headache events fulfilling the criteria of “probable migraine” (fair agreement), and 0.13 (SD 0.23) for events scored as “not migraine” (poor agreement).

### 3.3. Similarity of Headache Events

The median of the per-participant average IoU headache event similarity for the total set of headache events was 24.4% (Q1–Q3 18.5–33.7) (Table 4). When analyzing intraparticipant headache event pairs and comparing them by their assigned group labels, a disparity for the ICHD-3 symptom IoU similarity is observed. Event pairs labeled as “definite migraine” demonstrate a substantially higher ICHD-3 IoU similarity (52.6%) in comparison with those labeled as “probable migraine” (30.5%, *p* < 0.0001) and “not migraine” (14.6%, *p* < 0.0001), as indicated in Table 4 and Figure 5A. Event pairs of the “not migraine” group label exhibit the overall lowest IoU similarity for the ICHD-3 symptom list (Table 4). Note that Table 4 and Figure 5A–C do not align perfectly, as Table 4 does not utilize any form of binning or cutoff.

The ICHD-3 symptom list similarity trend analysis, illustrated by Figure 5D, indicates a decreasing yet not significant trend for both the “probable migraine” and “definite migraine” label groups, considering the 95% CI.

### 3.4. New ICHD-3 Symptoms Registered, Overreporting, and Underreporting

More than half of participants entered at least one new ICHD-3 symptom (15/27, 55.6%). The median number of new ICHD-3 symptoms registered during the study was one (Q1–Q3 0–3). Two participants registered five new ICHD-3 symptoms (7.4%) and two participants recorded six new ICHD-3 symptoms (7.4%) (Figure 6). A full overview of newly registered ICHD-3 symptoms is presented in Table 5. Appendix A provide the mean reporting ratios, along with the mean reporting once ratios, for the different migraine label groups.

## 4. Discussion

This study is one of the largest continuous EMA-based studies investigating the longitudinal variability of patient-specific headache event symptomatology in patients with migraine by using a dedicated digital smartphone headache diary.

An important first observation is that, in our cohort of participants diagnosed with migraine according to ICHD-3 criteria, more than 85% of headache events registered (429/505) did not meet the formal criteria within the ICHD-3 for migraine (as per the criteria within ICHD-3 1.1, migraine without aura). The purpose of the ICHD-3 criteria lies in making a diagnosis of the disorder from a patient’s history. Nevertheless, as our data show, clinicians need to be aware that most headache events in patients with migraine do not align with the full set of criteria, mostly because of the wide spectrum of the associated symptomatology, location, duration, and intensity of individual headache events. One specific note here needs to be made that most participants used adequate acute and preventive treatments. For the ICHD-3 criteria, we also excluded the criterion of untreated and/or unsuccessfully treated attacks due to the real-world setting of our study. As such, the effectiveness and variability in the use of acute self-medication may contribute towards the observed variability in headache events’ characteristics, and additionally leading to a lower incidence of headaches classified as “definite migraine” in the study.

On the basis of previous literature and clinical experience, our hypothesis was that moderate intake symptom agreement and low to moderate intraparticipant event similarity would be present in our dataset. The results of our study confirm that headache phenotyping during clinical interviews addresses the full syndrome at large (Table 2 and Table 3) and that both the agreement and similarity are rather low, indicating high symptom variability, as expected (Figure 4 and Figure 5).

Our analysis indicates that symptomatology for “definite migraine” headache events is, on average, slightly more than 50% in agreement with the migraine symptomatology presented during clinician-led intake interviews (Figure 4C). The “probable migraine” and “not migraine” group labels demonstrate a significantly lower agreement compared with “definite migraine”, but still a median symptomatology agreement of 40% exists for events with the “not migraine” group label. Consequently, “definite migraine” attacks demonstrate moderate agreement using Cohen’s kappa analysis, whereas “probable migraine” and “not migraine” events yield fair to low agreement, respectively. This highest rate of agreement for the “definite migraine” group label aligns with our hypothesis, yet it is lower than anticipated, considering the uniform method of registration and the phenotyping of the headache syndrome during clinical interviews. This discrepancy likely stems from a true form of variability of headache event symptomatology.

In the analysis of intake agreement over study time, we observe minimal temporal agreement variation (considering the 95% CI). This could reflect a true observation; however, three other possible explanations should be considered: (i) the sample size has selection bias or may be numerically insufficient, (ii) the 90-day study period could be too brief to detect changes in agreement, or (iii) the persistently low levels of agreement observed might inhibit the detection of noticeable changes in this type of analysis, a possibility suggested by the outcomes of the Cohen’s kappa agreement analyses.

Both the “definite migraine” and “probable migraine” label groups have a higher number of reported ICHD-3 and non-ICHD-3 symptoms, as shown in Figure 4G–I. The increased reporting of ICHD-3 symptoms can be linked to the requirement of specific ICHD-3 symptoms for the classification as “probable” or “definite” migraine, while non-ICHD-3 symptoms are excluded from these classification criteria. Note that the agreement of headache events for the non-ICHD-3 symptom subset is entirely determined by the event’s number of selected non-ICHD-3 symptoms, since only non-ICHD-3 symptoms that were marked as “present” during the intake are available within the longitudinal phase of the study. Additionally, the symptom occurrence subplots for the “definite migraine” label group display a V-shape, likely reflecting patient compliance and adherence throughout the study.

In alignment with the agreement results, the binned IoU similarity of the “definite migraine” label group is statistically higher than that for the “probable migraine” (*p* < 0.001) and “not migraine” (*p* < 0.0001) label groups (see Figure 5C). The IoU similarity trend analysis indicates that headache events become less similar as the time between them increases, aligning with our hypothesis. However, this trend appears to be not significant, considering the 95% confidence bounds. Furthermore, as illustrated in Figure 5A–C and Table 4, there is higher IoU similarity for the ICHD-3 symptom subgroup compared with the non-ICHD-3 subgroup in both the “definite migraine” and “probable migraine” attack pairs. In contrast, the “not migraine” subgroup shows lower ICHD-3 symptom IoU similarity than the non-ICHD-3 subgroup, likely influenced by the requirement of specific ICHD-3 symptoms for classification as “definite” or “probable” migraine.

Lastly, our results reveal an evolution in the range of ICHD-3 symptoms associated with headaches, as observed by analyzing newly identified ICHD-3 symptoms during the longitudinal registration phase (Figure 6). Notably, the emergence of new ICHD-3 symptoms was more common than rare, with some patients reporting up to five or six additional symptoms. It is interesting to note that multiple ICHD-3 symptoms within the criteria for cluster headache have been reported longitudinally in patients diagnosed with migraine and who did not register the symptoms during intake, underscoring the importance of regularly updating diagnostic criteria such as the ICHD-3 to accurately take into consideration the breadth of clinical presentations. For example, rhinorrhea, nasal congestion, and restlessness/agitation were new symptoms in over one-third of participants who denied the symptoms’ presence during intake. The most overreported symptoms seem to be vomiting, conjunctival injection, and lacrimation.

If we compare our research with previous literature on longitudinal evaluations of headache symptomatology, our results confirm the results by the group of Russell, who found that patients presented a more enriched symptomatology during the clinical interview in comparison with the longitudinally registered headache events in paper-based headache diaries [7]. Andrasik et al. also reported on the low concurrent validity between questionnaires and daily headache recordings in patients with tension-type headache [30]. Nachit-Ouinke et al. studied the reproducibility of the formal diagnosis of migraine in a cohort of headache sufferers by sending two postal questionnaires concerning headache symptoms and features at 12-monthly intervals [20]. They found that the one-year reproducibility of reporting migraine headache symptoms is only moderate, varies between symptoms, and leads to instability in the formal assignment of a migraine headache diagnosis and to diagnostic drift between headache types, aligning with our agreement analyses. In a group of unselected patients with migraine, Lieba-Samal et al. found days with the typical ICHD-3 symptoms of nausea/vomiting, photophobia, and phonophobia to be stable over the course of 30 days, aligning with our findings available in Appendix A [31]. Viana et al. looked at the similarity of three consecutive migraine headache events for the following characteristics: pain intensity, presence of nausea, vomiting, photophobia, phonophobia, osmophobia, allodynia, cranial autonomic symptoms (at least one), and premonitory symptoms [6]. By using a prospective diary-based study, the authors demonstrated that migraine headache events can be rather different, not just among patients but also in the same patient. They found none of the patients presented identical characteristics for the three studied headache events.

Our study highlights the complexity and intrapatient variability of headache symptomatology and underscores the importance of regularly evaluating and updating diagnostic criteria such as the ICHD-3 to accurately capture the breadth of clinical presentations. ICHD-3 has great clinical value for the diagnosis of headache disorders but, as our study shows, is limited when it comes to the follow-up of headache disorders. Especially in situations where outcome parameters such as the number migraine days or attacks are expected (e.g., clinical trials or reimbursement schemes of drugs), the intraparticipant symptom variability can play an important role. The study also demonstrates the continuous longitudinal learning process for both patients and physicians regarding the clinical features of the disorder. Our analyses demonstrate that electronic headache diaries leverage the capabilities to analyze EMA headache event symptomatology. The benefits of smartphone-based headache diaries compared with paper-based headache diaries are manifold and include scalability, structured collection of timestamped data, flexibility in data collection, and the ability to perform advanced and fine-grained data analysis. Our methodological approach can be scaled towards larger research efforts in the future, which may deliver more robust insights not only from the general group of patients with migraine, but also from subgroups of patients with migraine (e.g., according to age, sex, or gender). Additionally, it could extend to patients with other headache disorders such as cluster headache or tension-type headache. Apart from the technological possibilities, our study also finds evidence for issues of compliance with electronic headache diaries, as the median amount of days when the questionnaires were entered was only 76%, with a wide range between 37.2 and 91.6% of days within the study. This important aspect should always be addressed in future studies with electronic headache diaries.

Limitations to our study need to be addressed. First, our cohort was rather small due to the technological (limited to Android users with adequate technical skills and dedication to this study) and clinical constraints (inclusion and exclusion criteria) in the recruitment and participation of study participants. Therefore, the results need to be validated in a larger longitudinal, multinational group of patients with migraine. Second, the ICHD-3 symptomatology, but not the non-ICHD-3 symptomatology, questioned during the intake was available for entry during the longitudinal phase of the study, limiting our non-ICHD-3 symptom analysis during the longitudinal registration period. Third, there may be a potential bias towards overreporting symptoms as "any phase" when using a structured questionnaire during intake, as opposed to a traditional patient–physician consultation (Table 2). Furthermore, prompting patients with symptoms may also lead to higher number of baseline symptoms, as was demonstrated in previous literature [32]. Fourth, this study examined a cohort of participants with migraine who had access to effective acute and preventive treatments, simulating real-world clinical settings. Fifth, as this was a longitudinal observational study on migraine symptoms, no control group was included within this study.

## 5. Conclusions

Our real-world longitudinal study results show that electronically registered headache events fulfilling the criteria for “migraine without aura” (i.e., “definite migraine” in our study) show statistically higher agreement with the symptomatology from clinician-led intake interviews compared with events not meeting the criteria for “migraine without aura”. Events fulfilling the criteria for “migraine without aura” also had the highest longitudinal intraparticipant event pair similarity compared with events not meeting the criteria, with a declining trend for events spread out more in time. Most patients with migraine registered at least one new ICHD-3 symptom along the course of continuous headache event registrations. Our scalable study design paves the way for future research to extend this methodology to larger cohorts for more robust findings.

## Figures and Tables

**Figure 1 neurolint-17-00033-f001:**
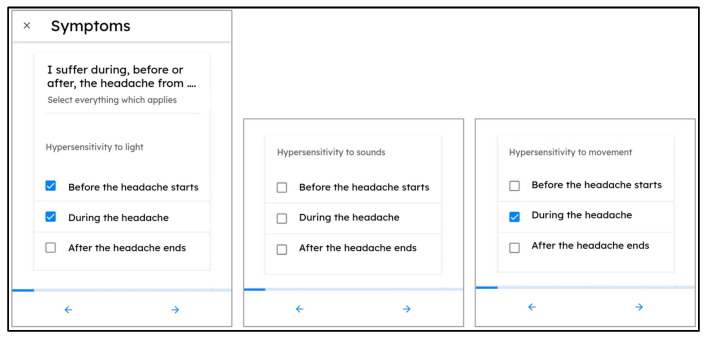
User interface for the submission of answers during the clinician-led intake interview. Blue tick boxes are checkboxes (multiple answers possible).

**Figure 2 neurolint-17-00033-f002:**
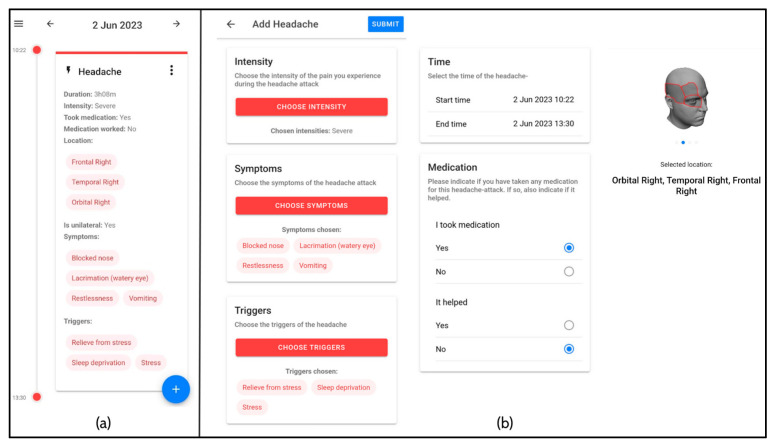
Headache registration interface for participants during the longitudinal phase of the study. (**a**) The overview of headache events registered in a timeline. (**b**) The headache registry module with different steps to be completed.

**Figure 3 neurolint-17-00033-f003:**
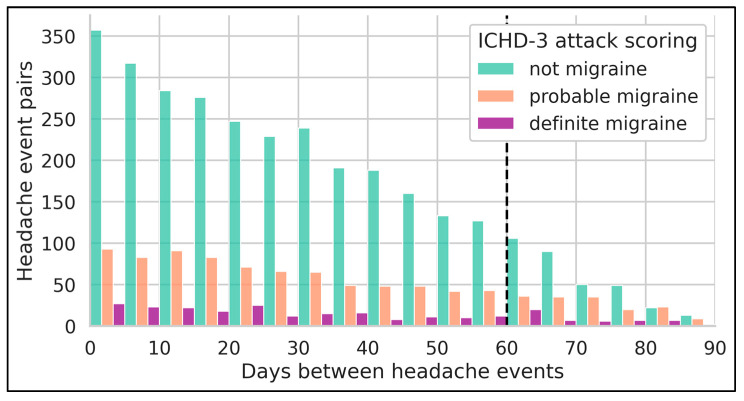
Overview of available headache event pairs per 5-day bin. The x-axis indicates the distance in days between the headache intervals of an event pair. Only events of the same participant with the same attack scoring can be paired. The vertical black dashed line indicates the upper headache-pair threshold.

**Figure 4 neurolint-17-00033-f004:**
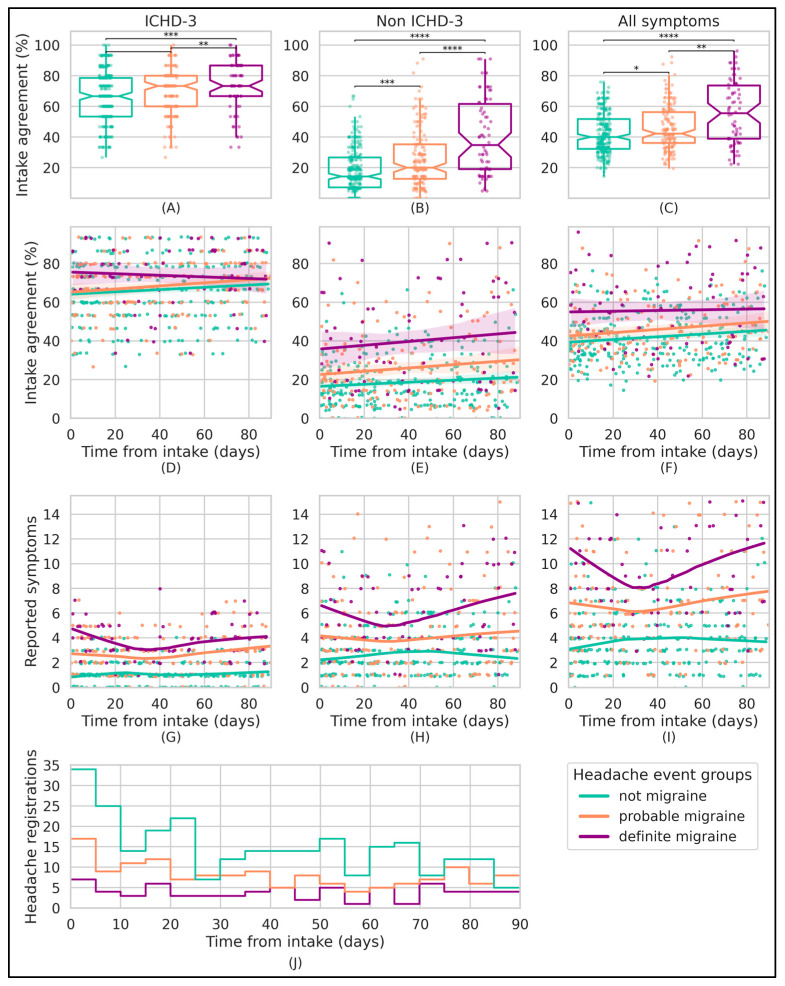
Headache event symptomatology agreement with clinician-led intake interviews and its trend over time. Each data point in the subplots corresponds to a bias-free headache event (*n* = 505), color-coded by the ICHD-3 headache event label. The subplots are divided into columns on the basis of the symptom subset, showing either agreement with the clinician-led intake interviews (Rows 1–2) or the number of checked symptoms (Row 3). Subplots (**A**–**C**) use box plots to depict the distribution of intake agreement for different subgroups, with statistical differences determined by the two-sided Mann–Whitney U test. Subplots (**D**–**F**) illustrate the trend of this intake agreement for the study period, with a first-order regression line and a 95% confidence interval for each headache event label group. The third row (subplots (**G**–**I**)), formatted similarly to the row above, displays the count of checked symptoms over time, but uses a locally weighted regression to emphasize local trends. Finally, the subplot (**J**) represents a histogram that counts the available headache events for each label group over time, using 5-day bins. Note: *p*-values: **** = *p* < 0.0001, *** = *p* < 0.001, ** = *p* < 0.01, * = *p* < 0.05. Abbreviations: ICHD-3, International Classification of Headache Disorders Third Edition; Q1, first quartile; Q3, third quartile.

**Figure 5 neurolint-17-00033-f005:**
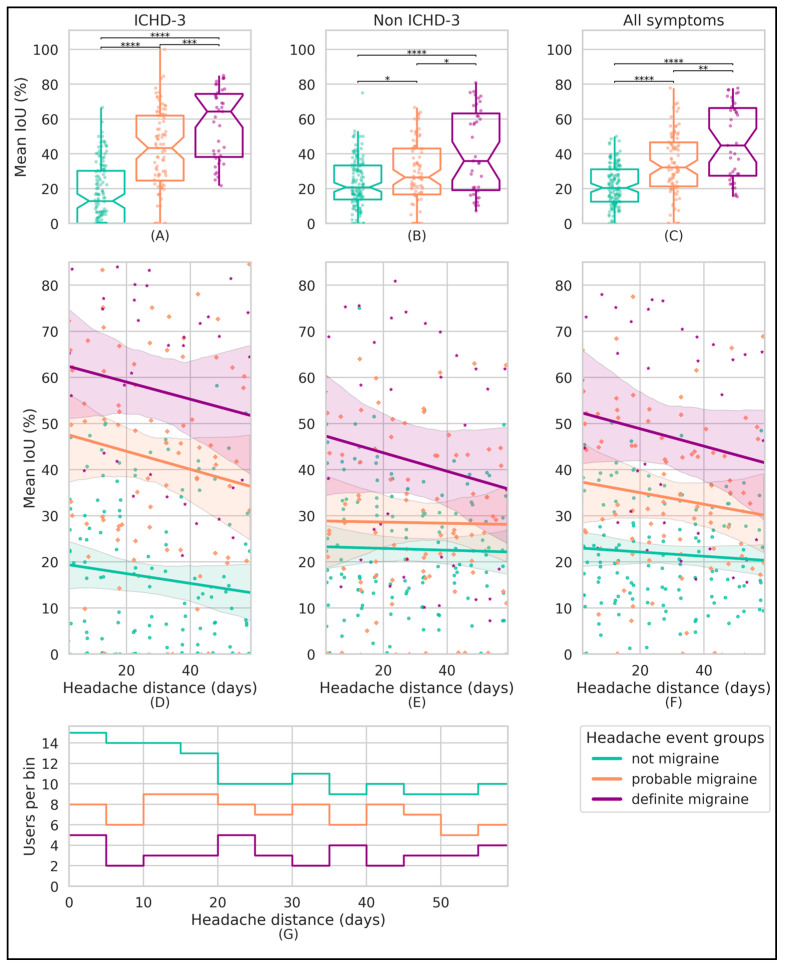
Per-participant mean IoU symptom similarity using paired headache event bins (bin size = 5 days). All subplots display median-aggregated IoU values of headache event pairs, using 5-day bins based on the headache distance of each pair. The headache event pairs were formed by combining headache events from each participant that have the same group label (i.e., “definite migraine”, “probable migraine”, or “not migraine”) and are color-coded accordingly. The columns of the first two subplot rows (subplots (**A**–**F**)) indicate the symptom sublists. In the first row (**A**–**C**), box plots illustrate the distribution of these per-participant mean-aggregated similarity values across different subgroups, with the two-sided Mann–Whitney U test indicating statistical differences. The second row (**D**–**F**) visualizes the similarity trend over time, represented by the distance of headache event pairs, featuring a first-order regression line and a 95% confidence interval for each classification group. The bottom subplot (**G**) is a histogram that counts the number of participants providing these mean aggregated values for each corresponding headache event label and 5-day bin. Note: *p*-values: **** = *p* < 0.0001, *** = *p* < 0.001, ** = *p* < 0.01, * = *p* < 0.05. Abbreviations: ICHD-3, International Classification of Headache Disorders, Third Edition; IoU, intersection over union.

**Figure 6 neurolint-17-00033-f006:**
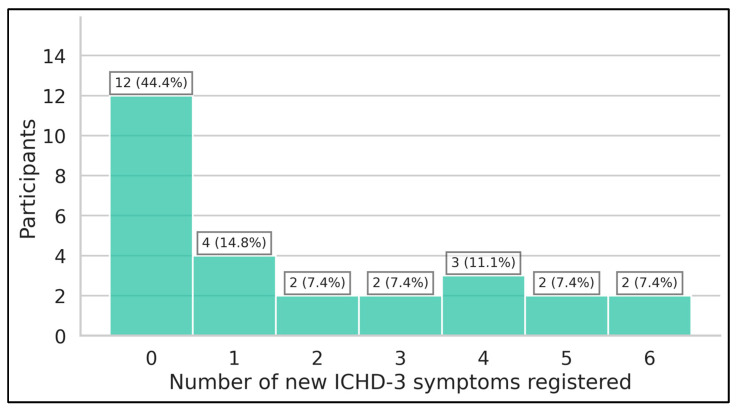
Number of participants registering new canonical ICHD-3 symptoms (*n* = 27).

**Table 1 neurolint-17-00033-t001:** Baseline characteristics of participants, *n* = 27.

Age, median (Q1–Q3)	35 (27–42)
Sex, *n* (%)	Female 21 (78%), Male 6 (22%)
Average migraine days per month, median (Q1–Q3)	6 (3–10)
Duration of headache syndrome in years, median (Q1–Q3)	17 (10–25)
Current use of acute treatment, *n* (%)	26 (96.3%)
Current use of preventive treatment, *n* (%)	15 (55.6%)
Days in study, median (Q1–Q3)	91 (83–97)
Days with the daily questionnaire completed	
(compliance rate, %), median (Q1–Q3)	76% (37.2–91.6%)

**Table 2 neurolint-17-00033-t002:** Description of headache events.

Headache Event Classification as per ICHD-3 Criteria	*n*	Number of Participants	Reported Symptoms per Headache Event, Median (Q1–Q3)/ Available Symptoms per Participant, Median (Q1–Q3)			Headache Event Duration in Hours and Minutes, Mean (SD)
			All Symptoms	ICHD-3	Non-ICHD-3	
**All headache events**	572	100% (27/27)	5 (3–8) 33 (32–37)	2 (1–3) 16 (*)	3 (2–5) 17 (16–21)	7 h 09 min (8 h 35 min)
**Eligible headache events**	505	100% (27/27)	5(3–8) 33 (32–37)	2 (1–3) 16 (*)	3 (2–5) 17 (16–21)	6 h 42 min (8 h 16 min)
**Not migraine**	276	92.6% (25/27)	3 (2–5) 32 (32–33)	1 (0–2) 16 (*)	2 (1–4) 16 (16–17)	4 h 28 min (5 h 27 min)
**Probable migraine**	155	77.8% (21/27)	6 (4–11) 34 (33–38)	3 (1–4) 16 (*)	4 (2–6) 18 (17–22)	8 h 33 min (9 h 16 min)
**Definite migraine**	74	63.0% (17/27)	9 (7–14) 34 (31–38)	4 (3–5) 16 (*)	5 (4–10) 18 (15–22)	11 h 08 min (11 h 32 min)

(*) All ICHD-3 symptoms were present during headache registration (resulting in no variation in symptom availability). Abbreviations: h, hour; ICHD-3, International Classification of Headache Disorders, Third Edition; n, number; Q1, first quartile; Q3, third quartile; SD, standard deviation.

**Table 3 neurolint-17-00033-t003:** Agreement between headache events and clinician-led intake interviews, grouped by symptoms’ presence during intake.

	Intake: Symptom is Present and Headache Event: Symptom Is Present	Intake: Symptom Is Not Present and Headache Event: Symptom Is Not Present
	**ICHD-3 symptoms**	
**All headache events**	20.0% (0.0–50.0)	100% (100–100) (mean: 96.9%)
**Not migraine**	10.6% (0.0–28.6)	100.0% (100–100) (mean: 97.2%)
**Probable migraine**	33.3% (14.3–50.0)	100.0% (100–100) (mean: 96.8%)
**Definite migraine**	53.6% (28.6–75.0)	100.0% (90.4–100) (mean: 95.7%)
	**Non-ICHD-3 symptoms**	
**All headache events**	18.75% (10.0–33.3)	/
**Not migraine**	14.3% (7.1–26.7)	/
**Probable migraine**	20.0% (12.5–34.2)	/
**Definite migraine**	34.8% (19–61.6)	/

Note: all results are median (Q1–Q3) unless otherwise specified in the table.

**Table 4 neurolint-17-00033-t004:** Median of intraparticipant mean IoU similarity of intraparticipant headache event pairs (without bin aggregation or cutoff after 60 days).

ICHD-3 Headache Event Classification	Number of Patients	Headache IoU Similarity (%) Median (Q1–Q3)		
		All Symptoms	ICHD-3	Non-ICHD-3
All headache events	24 (*)	24.4% (18.5–33.7)	22.5% (6.2–36.2)	25.8% (16.5–43.9)
Not migraine	21	20.8% (16.1–35.0)	14.6% (0.0–33.3)	21.7% (16.7–35.3)
Probable migraine	17	24.0% (18.1–40.7)	30.5% (16.8–45.3)	23.7% (19.3–33.3)
Definite migraine	12	37.6% (26.3–48.8)	52.6% (33.1–64.0)	32.2% (23.6–51.1)

(*) This number differs from Table 2, as patients need to have at least two events of an ICHD-3 event label to form pairs. Abbreviations: ICHD-3, International Classification of Headache Disorders, Third Edition; IoU, intersection over union; Q1, first quartile; Q3, third quartile.

**Table 5 neurolint-17-00033-t005:** Mean and median reporting ratio of ICHD-3 symptoms, grouped by user.

Symptom	Intake Value	Number of Participants/ Number of Headache Registrations		Mean Symptom Reporting Ratio (%) (SD)	Median Symptom Reporting Ratio (Q1–Q3)	Participant Ratio Who Reported the Symptom at Least Once
**Conjunctival injection**	Present	4	100	4.4% (8.8)	0.0% (0.0–4.4)	25.0% (1/4)
	Not present	23	405	0.7% (2.3)	0.0% (0.0–0.0)	13.0% (3/23)
**Eyelid edema**	Present	4	77	23.4% (25.3)	19.6% (4.4–38.6)	75.0% (3/4)
	Not present	23	428	1.2% (3.9)	0.0% (0.0–0.0)	13.0% (3/23)
**Forehead and facial sweating**	Present	9	210	9.1% (6.8)	8.3% (5.9–12.5)	77.8% (7/9)
	Not present	18	295	0.0% (0.0)	0.0% (0.0–0.0)	0.0% (0/18)
**Lacrimation**	Present	6	135	3.6% (5.6)	0.0% (0.0–7.5)	33.3% (2/6)
	Not present	21	370	3.0% (5.4)	0.0% (0.0–7.1)	28.6% (6/21)
**Miosis**	Present	2	21	0.0% (0.0)	0.0% (0.0–0.0)	0.0% (0/2)
	Not present	25	484	0.1% (0.4)	0.0% (0.0–0.0)	4.0% (1/25)
**Motion sensitivity**	Present	17	302	23.3% (27.6)	16.0% (0.0–35.3)	64.7% (11/17)
	Not present	10	203	2.6% (5.5)	0.0% (0.0–1.3)	30.0% (3/10)
**Nasal congestion**	Present	6	100	35.3% (33.0)	26.7% (11.6–45.8)	100.0% (6/6)
	Not present	21	405	7.3% (17.5)	0.0% (0.0–7.5)	38.1% (8/21)
**Nausea**	Present	23	406	35.1% (30.3)	26.5% (2.9–54.0)	73.9% (17/23)
	Not present	4	99	2.2% (2.6)	1.8% (0.0–4.0)	50.0% (2/4)
**Phonophobia**	Present	22	445	27.3% (31.0)	15.2% (0.0–46.9)	68.2% (15/22)
	Not present	5	60	0.0% (0.0)	0.0% (0.0–0.0)	0.0% (0/5)
**Photophobia**	Present	21	443	37.0% (36.1)	20.0% (0.0–73.5)	71.4% (15/21)
	Not present	6	62	1.0% (2.4)	0.0% (0.0–0.0)	16.7% (1/6)
**Ptosis**	Present	5	105	31.5% (39.0)	20.0% (12.0–22.5)	100.0% (5/5)
	Not present	22	400	0.3% (1.3)	0.0% (0.0–0.0)	4.5% (1/22)
**Restlessness or agitation**	Present	9	163	28.0% (23.4)	25.0% (10.0–35.3)	88.9% (8/9)
	Not present	18	342	2.8% (4.6)	0.0% (0.0–3.9)	33.3% (6/18)
**Rhinorrhea**	Present	7	132	11.2% (11.6)	11.8% (0.0–19.1)	57.1% (4/7)
	Not present	20	373	4.0% (6.6)	0.0% (0.0–5.0)	40.0% (8/20)
**Throbbing headache**	Present	17	288	17.8% (19.9)	10.7% (5.9–20.0)	76.5% (13/17)
	Not present	9	159	3.1% (5.5)	0.0% (0.0–2.5)	44.4% (4/9)
**Vomiting**	Present	12	206	7.1% (15.2)	0.0% (0.0–4.2)	33.3% (4/12)
	Not present	15	299	1.3% (4.0)	0.0% (0.0–0.0)	13.3% (2/15)

## Data Availability

The datasets used and analyzed during the current study are available from the corresponding author on reasonable request for academic research purposes only.

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
