# Peer review of "Tracking Migraine Symptoms: A Longitudinal Comparison of Smartphone-Based Headache Diaries and Clinical Interviews"

_2035-8377, 2025, doi:10.3390/neurolint17030033_

Round 1
Reviewer 1 Report
Comments and Suggestions for Authors
Thank you very much for this manuscript.
This is a very thorough presentation of longitudinally collected data in migraine, which is of high interest for the reader. Methods have been described to the detail, as well as results. However, some sections use highly elaborate sentences, which may pose challenges for the reader in following the flow of the argument. Simplifying these areas could enhance clarity and accessibility while maintaining the manuscript's scholarly rigor. Congratulations on producing such an excellent and valuable contribution
Author Response
Thank you very much for this manuscript.
This is a very thorough presentation of longitudinally collected data in migraine, which is of high interest for the reader. Methods have been described to the detail, as well as results. However, some sections use highly elaborate sentences, which may pose challenges for the reader in following the flow of the argument. Simplifying these areas could enhance clarity and accessibility while maintaining the manuscript's scholarly rigor. Congratulations on producing such an excellent and valuable contribution
Thank you for your review, and we appreciate the suggestion. We have fully reread the document and revised multiple sentence constructions which were deemed to be elaborate and complex for the reader. We hope to have improved the readability of the manuscript to the reviewer’s satisfaction.
Reviewer 2 Report
Comments and Suggestions for Authors
Dear authors,
Congratulations on submitting your paper for publication. The record of headache outcomes play a pivot role in the management of headache, therefore your articles may be of interest for a broad professionals involved in the study and in management of headache. The following suggestions could improve the quality of your research.
Abstract
1. please try to make your abstract more attractive, add summarize the main findings and report in the results some statistical significant differences found
Introduction
2. The burden of migraine the need for innovative technological management should described to more explicitly connect the study's potential impact to broader clinical and societal implications.
3. the pathophysiology of migraine should be taken in to consideration
4. I suggest the following structures: first paragraph the burden and pathophysiology of migraine, second paragraph the importance to record headache outcomes and the most common method to record it, third the smartphone-based headache diary, fourth the primary and secondary outcomes and the hypotheses. This structure lead to a more congruent in flow of ideas, and continuity of thought and it would enhance the reader's understanding of the study's direction.
Method
5. the methods should report more details regarding: chosen sample size, institutional review board, control group
Discussion
6. I suggest to add a paragraph concerning the suggestions of general hypotheses concerning the observed benefits of smartphone-based headache diary
7. More thorough consideration of the study's generalizability to different headache populations and settings would help readers assess the findings' applicability.
Author Response
Reviewer 2
Dear authors,
Congratulations on submitting your paper for publication. The record of headache outcomes play a pivot role in the management of headache, therefore your articles may be of interest for a broad professionals involved in the study and in management of headache. The following suggestions could improve the quality of your research.
Abstract
- please try to make your abstract more attractive, add summarize the main findings and report in the results some statistical significant differences found
Thank you for this remark. We adapted the abstract with a more concise flow of words and sentences providing better readability. The main findings are summarized and present the p-values from the statistical testing.
|
Abstract: Background/Objectives: By leveraging the capabilities of a smartphone-based headache diary, the objective of the study is to determine the amount of agreement of mi-graine-associated symptomatology during headache events and the symptoms docu-mented during clinician-led intake interviews. Methods: This was a 90-days longitudinal, smartphone-based headache calendar study for participants diagnosed with migraine. Registered headache events were labeled as “definite migraine”, “probable migraine” and “not migraine” in accordance with International Classification of Headache Disorders, Third Edition (ICHD-3) criteria. Symptom-agreement with clinician-led intake interviews (agreement percentages and kappa-coefficients), symptom-similarity between headache events within users (percentage) and amount of newly registered ICHD-3 symptoms per participant were calculated. Results: Twenty-seven participants provided 505 headache events eligible for analysis. The median agreement between recorded headache event symptomatology and clinician-led intake interview phenotyping ranged between 40% (for events fulfilling “not migraine” criteria) and 55.5% (“definite migraine”) (p < 0.001). Higher intraparticipant headache event pair similarity was observed for “definite mi-graine” pairs (p < 0.01), along with a decreasing trend in similarity as attack-pair head-ache distance increases. Over half of the participants registered at least one new ICHD-3 symptom during the study. Conclusions: Electronic diary registrations show substantial longitudinal variability in intrapersonal headache symptomatology, with similarity of headache events declining over time. The registration of a new ICHD-3 symptom was the rule rather than the exception. |
Introduction
- The burden of migraine the need for innovative technological management should described to more explicitly connect the study's potential impact to broader clinical and societal implications.
- The pathophysiology of migraine should be taken in to consideration
- I suggest the following structures: first paragraph the burden and pathophysiology of migraine, second paragraph the importance to record headache outcomes and the most common method to record it, third the smartphone-based headache diary, fourth the primary and secondary outcomes and the hypotheses. This structure lead to a more congruent in flow of ideas, and continuity of thought and it would enhance the reader's understanding of the study's direction.
We appreciate the elaborate reflection on the structure of the introduction within points 2, 3 and 4 stated above. We have considered all requests and suggestions. Our efforts led to the adaptation of the introduction in the revised manuscript including the renewed structuring as suggested above. We would like to refer to the revised manuscript for the full reading of the introduction.
Method
- the methods should report more details regarding: chosen sample size, institutional review board, control group
We thank you for your kind remarks regarding the methodology section. We provide the requested information below and have adapted the manuscript accordingly.
The presented study was part of the “mBrain” study design. “mBrain” was a comprehensive and longitudinal observational research initiative as detailed in De Brouwer et al.[1] enrolling participants with migraine. The sample size was therefore not calculated a priori.
|
The study design did not preset a fixed sample size nor did it calculate a sample size through power calculation. The “mBrain” study was an observational pilot study where the aim was to include as many participants with migraine as possible. |
As to whether we recruited a control group, we have added the following sentence under subheading 2.2 “Study Design”:
|
As this study only investigates events related to migraine attacks, a control group of participants without migraine was not recruited. |
The institutional board review was already mentioned under subheading 2.5 “Ethics approval and patients’ consent”:
|
2.5. Ethics approval and patients’ consent The study was approved by the Ethics Committee of University Hospital Ghent (BC-10031). The study was preregistered at clinicaltrials.gov (NCT04983186). All patients gave informed consent for the collection and analysis of their pseudonymized data, and for publication of the results. |
Discussion
- I suggest to add a paragraph concerning the suggestions of general hypotheses concerning the observed benefits of smartphone-based headache diary
Thank you for this valuable remark. We have added the following sentence in the discussion:
|
The benefits of smartphone-based headache diaries compared to paper-based headache diaries are manifold and include scalability, structured collection of timestamped data, flexibility in data collection and the ability to perform advanced and fine-grained data analysis. |
- More thorough consideration of the study's generalizability to different headache populations and settings would help readers assess the findings' applicability.
We acknowledge the necessity of discussing the generalizability of the study findings in the discussion section. Therefore, we have further expanded this section in the discussion as follows:
|
Our methodological approach can be scaled towards larger research efforts in the future, which may deliver more robust insights not only from the general group of patients with migraine, but also from subgroups of patients with migraine (e.g., according to age, sex, or gender). Additionally, it could extend to patients with other headache disorders such as cluster headache or tension-type headache. Apart from the technological possibilities, our study also finds evidence for issues of compliance with electronic headache diaries as the median number of days where the questionnaires were entered—which was utilized to assess compliance—was only 76% with a wide range between 37.2 and 91.6 % of days within the study. This important aspect should always be addressed in future studies with electronic headache diaries. |
[1] De Brouwer, M.; Vandenbussche, N.; Steenwinckel, B.; Stojchevska, M.; Van Der Donckt, J.; Degraeve, V.; Vaneessen, J.; De Turck, F.; Volckaert, B.; Boon, P.; et al. mBrain: Towards the Continuous Follow-up and Headache Classification of Primary Headache Disorder Patients. BMC Med. Inform. Decis. Mak. 2022, 22, 87, doi:10.1186/s12911-022-01813-w.
Reviewer 3 Report
Comments and Suggestions for Authors
Dear authors,
First of all thank you for the invitation to review your study “A longitudinal comparative study of headache event symptomatology in patients with migraine between smartphone-based
headache diary entries and clinician-led intake interviews”.
Please find some specific comments below
INTRODUCTION
- The introduction need to better described the burden of migraine and their complex physiopathology, in order to support your work. In fact, the burden of migraine is also due to their complex and interesting physiopathology that involve two opposing processes lack of habituation and sensitization that, together, they lead to dysfunction in cortical excitability and pain processing. Please take in to consideration this two recent articles, the first on physiopathology of migraine the second one on the burden of migraine:
Deodato M, Granato A, Martini M, Buoite Stella A, Galmonte A, Murena L, Manganotti P. Neurophysiological and Clinical Outcomes in Episodic Migraine Without Aura: A Cross-Sectional Study. J Clin Neurophysiol. 2024 May 1;41(4):388-395. doi: 10.1097/WNP.0000000000001055. PMID: 37934069.
GBD 2016 Headache Collaborators. Global, regional, and national burden of migraine and tension-type headache, 1990-2016: a systematic analysis for the Global Burden of Disease Study 2016. Lancet Neurol 2018;17:954–976.
- Please delineate more precisely the specific research gap that your study aims to fill, particularly in the context of existing methods to record migraine parameters such as frequency of migraine, duration of attack, pain intensity, number of drug intake, menstrual cycle, sleep…there are important parameters for a clinical assessment of the efficacy of pharmacological and non-pharmacological treatments. In this line to find a smarter systems to record it represents a challenge in the managements of migraine, but also for other type of headache.
METHOD
- Was the study approved by an institutional review board?
- The manuscript does not clearly justify the chosen sample size or discuss how it ensures sufficient power to detect expected results
- Inclusion/exclusion criteria should be better described, for example were patients that still underwent prophylactic pharmacological and non-pharmacological treatments included?
- Why did you not compare the record of this electronic diary with the standard method?
- Please add more references that support the use of digital tools to record patients outcomes in order to justify your method
RESULTS
- The quality of tables and figures are very good, nevertheless for the small sample it would be more appropriate to use medians and interquartile instead of standard deviation
DISCUSSION
- It could be interesting to discuss about the compliance of the patients to record these parameters in this electronic diary respect to a standard method, and discuss concerning the use of digital tools in medicine and in recording outocomes
- The discussion would benefit from more detailed suggestions for future research, including potential benefit in recording these headache parameters in order to better monitoring the effect of different treatments and to enhance the compliance of patients.
- While some limitations are acknowledged, a more comprehensive discussion of the study's limitations, including the small sample size but also the absence of a control group would provide a more balanced view.
Author Response
Reviewer 3
Dear authors,
First of all thank you for the invitation to review your study “A longitudinal comparative study of headache event symptomatology in patients with migraine between smartphone-based headache diary entries and clinician-led intake interviews”. Please find some specific comments below
INTRODUCTION
- The introduction need to better described the burden of migraine and their complex physiopathology, in order to support your work. In fact, the burden of migraine is also due to their complex and interesting physiopathology that involve two opposing processes lack of habituation and sensitization that, together, they lead to dysfunction in cortical excitability and pain processing. Please take in to consideration this two recent articles, the first on physiopathology of migraine the second one on the burden of migraine:
Deodato M, Granato A, Martini M, Buoite Stella A, Galmonte A, Murena L, Manganotti P. Neurophysiological and Clinical Outcomes in Episodic Migraine Without Aura: A Cross-Sectional Study. J Clin Neurophysiol. 2024 May 1;41(4):388-395. doi: 10.1097/WNP.0000000000001055. PMID: 37934069.
GBD 2016 Headache Collaborators. Global, regional, and national burden of migraine and tension-type headache, 1990-2016: a systematic analysis for the Global Burden of Disease Study 2016. Lancet Neurol 2018;17:954–976.
We acknowledge the fact that the pathophysiology of migraine has received little attention in our first submission. Therefore, we have added the current state of knowledge on the pathophysiology of migraine in the introduction, citing both suggested key review papers of recent years.
|
Migraine is a neurological disorder involving the trigeminovascular system and various central networks of the brain. The pathophysiology involves disturbances in sensory processing wherein lack of habituation to stimuli and sensitization of various neural structures play key roles. |
- Please delineate more precisely the specific research gap that your study aims to fill, particularly in the context of existing methods to record migraine parameters such as frequency of migraine, duration of attack, pain intensity, number of drug intake, menstrual cycle, sleep…there are important parameters for a clinical assessment of the efficacy of pharmacological and non-pharmacological treatments. In this line to find a smarter systems to record it represents a challenge in the managements of migraine, but also for other type of headache.
We fully agree with this remark and have adapted the text in the introduction to answer the question which research gaps remain within the field of smartphone-based headache diaries. The research gap this study aims to overcome is to understand how much heterogeneity and variability of intraparticipant migraine symptomatology exists over time when longitudinally registered in smartphone-based headache diaries. We have expanded this in the introduction as follows:
|
Previous studies have looked at the characteristics of singular headache events in individual patients and the evolution of headache event symptomatology through time [4,13]. These analyses are subject to undertaking a comprehensive investigation of several facets, including the comparison of symptomatology across distinct headache events, the alignment of headache event-related symptomatology with data collected during clinician-led headache intake interviews, and the longitudinal tracking of newly discovered symptoms by the participants [4,13]. The largest study to date by Verhagen et al. compared differences in migraine attack characteristics between men and women longitudinally through electronic headache diaries and found that compared to attacks in men, both perimenstrual and non-perimenstrual migraine attacks are of longer duration and are more often accompanied by associated symptoms. This study shows the capability of electronic headache diary studies to find latent aspects of migraine symptomatology. However, there still remains a research gap on the heterogeneity and variability of intraparticipant extended migraine symptomatology over time from longitudinal studies that apply smartphone-based headache diaries. |
METHOD
- Was the study approved by an institutional review board?
The institutional board review was already mentioned under subheading 2.5 “Ethics approval and patients’ consent”:
|
2.5. Ethics approval and patients’ consent The study was approved by the Ethics Committee of University Hospital Ghent (BC-10031). The study was preregistered at clinicaltrials.gov (NCT04983186). All patients gave informed consent for the collection and analysis of their pseudonymized data, and for publication of the results. |
- The manuscript does not clearly justify the chosen sample size or discuss how it ensures sufficient power to detect expected results
Thank you for this remark. The presented study was part of the “mBrain” study design. “mBrain” was a comprehensive and longitudinal observational research initiative as detailed in De Brouwer et al. enrolling participants with migraine . The sample size was therefore not calculated a priori. We have added this section to the methods:
|
The study design did not preset a fixed sample size nor did it calculate a sample size through power calculation. The “mBrain” study was an observational pilot study where the aim was to include as many participants with migraine as possible. |
- Inclusion/exclusion criteria should be better described, for example were patients that still underwent prophylactic pharmacological and non-pharmacological treatments included?
We agree with this valid remark that this has not been stipulated fully within the manuscript. We have added the following information under subheading 2.1 “Participants”:
|
Participants were allowed to use acute or preventive treatment as required from a clinical perspective with no plan to switch preventive treatment during the course of the study. |
- Why did you not compare the record of this electronic diary with the standard method?
Thank you for your question which we will explain. Although your suggestion would also result in interesting research questions, the intention of the study was not to compare electronic headache diaries with non-digital alternatives, but rather to investigate the real-world application of these electronic headache diaries and to understand the heterogeneity and variability of intraparticipant migraine symptomatology over time.
- Please add more references that support the use of digital tools to record patients outcomes in order to justify your method
Thank you for this remark. In the introduction section, we have added extra references from the literature on electronic headache diaries stipulating beneficial use cases for this methodology, building on the foundations of our research efforts.
Daniëls, Naomi E M, Laura M J Hochstenbach, Catherine Van Zelst, Marloes A Van Bokhoven, Philippe A E G Delespaul, en Anna J H M Beurskens. ‘Factors That Influence the Use of Electronic Diaries in Health Care: Scoping Review’. JMIR mHealth and uHealth 9, nr. 6 (1 juni 2021): e19536. https://doi.org/10.2196/19536.
Iribarren, Sarah J, Tokunbo O Akande, Kendra J Kamp, Dwight Barry, Yazan G Kader, en Elizabeth Suelzer. ‘Effectiveness of Mobile Apps to Promote Health and Manage Disease: Systematic Review and Meta-Analysis of Randomized Controlled Trials’. JMIR mHealth and uHealth 9, nr. 1 (11 januari 2021): e21563. https://doi.org/10.2196/21563.
Sim, Ida. ‘Mobile Devices and Health’. New England Journal of Medicine 381, nr. 10 (5 september 2019): 956-68. https://doi.org/10.1056/NEJMra1806949.
These references have been cited after the following new sentence in the introduction:
|
Electronic diaries are being implemented widely in medicine [1-3]. In headache medicine, it is worth noting that certain studies have indicated a patient preference for electronic headache diaries over traditional paper-based methods …
[1] Daniëls, Naomi E M, Laura M J Hochstenbach, Catherine Van Zelst, Marloes A Van Bokhoven, Philippe A E G Delespaul, en Anna J H M Beurskens. ‘Factors That Influence the Use of Electronic Diaries in Health Care: Scoping Review’. JMIR mHealth and uHealth 9, nr. 6 (1 juni 2021): e19536. https://doi.org/10.2196/19536. [2] Iribarren, Sarah J, Tokunbo O Akande, Kendra J Kamp, Dwight Barry, Yazan G Kader, en Elizabeth Suelzer. ‘Effectiveness of Mobile Apps to Promote Health and Manage Disease: Systematic Review and Meta-Analysis of Randomized Controlled Trials’. JMIR mHealth and uHealth 9, nr. 1 (11 januari 2021): e21563. https://doi.org/10.2196/21563. [3] Sim, Ida. ‘Mobile Devices and Health’. New England Journal of Medicine 381, nr. 10 (5 september 2019): 956-68. https://doi.org/10.1056/NEJMra1806949. Please note: reference numbers used in this box do not overlap with reference numbers used in the manuscript. |
RESULTS
- The quality of tables and figures are very good, nevertheless for the small sample it would be more appropriate to use medians and interquartile instead of standard deviation
Thank you for your valuable remark. We agree that, given the relatively small sample size, the data distribution is better represented using medians and interquartile ranges (IQRs). Accordingly, we have updated Table 1, as it presents descriptive statistics per participant (n=27).
For Tables 2, 3, and 4, we did not make any modifications, as the data was already expressed using medians and IQRs where appropriate, or we believe the sample size was sufficiently large to justify the use of means and standard deviations (SDs).
Regarding Table 5, we acknowledge that the mean reporting ratio could also be presented using medians and IQRs. Therefore, we have included these values as an additional reference.
As for the figures, we believe no changes are necessary, as they do not display means or SDs. The distributions are already effectively visualized using box plots.
Table 5. Mean and median reporting ratio of ICHD-3 symptoms, grouped by user.
|
Symptom |
Intake value |
Number of participants / |
Mean symptom reporting ratio (%) (SD) |
Median symptom reporting ratio (Q1-Q3) |
Participant ratio who reported the symptom at least once |
|
|
Conjunctival injection |
Present |
4 |
100 |
4.4% (8.8) |
0.0% (0.0 - 4.4) |
25.0% (1/4) |
|
Not present |
23 |
405 |
0.7% (2.3) |
0.0% (0.0 - 0.0) |
13.0% (3/23) |
|
|
Eyelid oedema
|
Present |
4 |
77 |
23.4% (25.3) |
19.6% (4.4 - 38.6) |
75.0% (3/4) |
|
Not present |
23 |
428 |
1.2% (3.9) |
0.0% (0.0 - 0.0) |
13.0% (3/23) |
|
|
Forehead and facial sweating |
Present |
9 |
210 |
9.1% (6.8) |
8.3% (5.9 - 12.5) |
77.8% (7/9) |
|
Not present |
18 |
295 |
0.0% (0.0) |
0.0% (0.0 - 0.0) |
0.0% (0/18) |
|
|
Lacrimation
|
Present |
6 |
135 |
3.6% (5.6) |
0.0% (0.0 - 7.5) |
33.3% (2/6) |
|
Not present |
21 |
370 |
3.0% (5.4) |
0.0% (0.0 - 7.1) |
28.6% (6/21) |
|
|
Miosis
|
Present |
2 |
21 |
0.0% (0.0) |
0.0% (0.0 - 0.0) |
0.0% (0/2) |
|
Not present |
25 |
484 |
0.1% (0.4) |
0.0% (0.0 - 0.0) |
4.0% (1/25) |
|
|
Motion sensitivity |
Present |
17 |
302 |
23.3% (27.6) |
16.0% (0.0 - 35.3) |
64.7% (11/17) |
|
Not present |
10 |
203 |
2.6% (5.5) |
0.0% (0.0 - 1.3) |
30.0% (3/10) |
|
|
Nasal congestion |
Present |
6 |
100 |
35.3% (33.0) |
26.7% (11.6 - 45.8) |
100.0% (6/6) |
|
Not present |
21 |
405 |
7.3% (17.5) |
0.0% (0.0 - 7.5) |
38.1% (8/21) |
|
|
Nausea
|
Present |
23 |
406 |
35.1% (30.3) |
26.5% (2.9 - 54.0) |
73.9% (17/23) |
|
Not present |
4 |
99 |
2.2% (2.6) |
1.8% (0.0 - 4.0) |
50.0% (2/4) |
|
|
Phonophobia
|
Present |
22 |
445 |
27.3% (31.0) |
15.2% (0.0 - 46.9) |
68.2% (15/22) |
|
Not present |
5 |
60 |
0.0% (0.0) |
0.0% (0.0 - 0.0) |
0.0% (0/5) |
|
|
Photophobia
|
Present |
21 |
443 |
37.0% (36.1) |
20.0% (0.0 - 73.5) |
71.4% (15/21) |
|
Not present |
6 |
62 |
1.0% (2.4) |
0.0% (0.0 - 0.0) |
16.7% (1/6) |
|
|
Ptosis
|
Present |
5 |
105 |
31.5% (39.0) |
20.0% (12.0 - 22.5) |
100.0% (5/5) |
|
Not present |
22 |
400 |
0.3% (1.3) |
0.0% (0.0 - 0.0) |
4.5% (1/22) |
|
|
Restlessness or agitation |
Present |
9 |
163 |
28.0% (23.4) |
25.0% (10.0 - 35.3) |
88.9% (8/9) |
|
Not present |
18 |
342 |
2.8% (4.6) |
0.0% (0.0 - 3.9) |
33.3% (6/18) |
|
|
Rhinorrhea
|
Present |
7 |
132 |
11.2% (11.6) |
11.8% (0.0 - 19.1) |
57.1% (4/7) |
|
Not present |
20 |
373 |
4.0% (6.6) |
0.0% (0.0 - 5.0) |
40.0% (8/20) |
|
|
Throbbing headache |
Present |
17 |
288 |
17.8% (19.9) |
10.7% (5.9 - 20.0) |
76.5% (13/17) |
|
Not present |
9 |
159 |
3.1% (5.5) |
0.0% (0.0 - 2.5) |
44.4% (4/9) |
|
|
Vomiting
|
Present |
12 |
206 |
7.1% (15.2) |
0.0% (0.0 - 4.2) |
33.3% (4/12) |
|
Not present |
15 |
299 |
1.3% (4.0) |
0.0% (0.0 - 0.0) |
13.3% (2/15) |
|
DISCUSSION
- It could be interesting to discuss about the compliance of the patients to record these parameters in this electronic diary respect to a standard method, and discuss concerning the use of digital tools in medicine and in recording outocomes
Thank you for this valid remark. We have highlighted this in the discussion section of our new manuscript.
|
… Our analyses demonstrate that electronic headache diaries leverage the capabilities to analyze EMA headache event symptomatology. Our methodological approach can be scaled towards larger research efforts in the future, which may deliver more robust insights not only from the general group of patients with migraine, but also from subgroups of patients with migraine (e.g., according to age, sex, or gender). Apart from the technological possibilities, our study also finds evidence for issues of compliance with electronic headache diaries as the median amount of days where the questionnaires were entered was only 76% with a wide range between 37.2 and 91.6 % of days within the study. This important aspect should always be addressed in future studies with electronic headache diaries. … |
- The discussion would benefit from more detailed suggestions for future research, including potential benefit in recording these headache parameters in order to better monitoring the effect of different treatments and to enhance the compliance of patients.
Thank you for this remark. We have added the following sentence in the discussion:
|
The benefits of smartphone-based headache diaries compared to paper-based headache diaries are manifold and include scalability, structured collection of timestamped data, flexibility in data collection and the ability to perform advanced and fine-grained data analysis. |
- While some limitations are acknowledged, a more comprehensive discussion of the study's limitations, including the small sample size, but also the absence of a control group would provide a more balanced view.
We acknowledge that our research described in the manuscript is subjected to many limitations with need to be addressed to balance the view on the results.
Regarding the sample size, this was addressed in the manuscript within the discussion section. The manuscript already describes the reasons for the small sample size. The text reads as follows:
|
First, our cohort was rather small due to the technological (limited to Android users with adequate technical skills and dedication to the study) and clinical constraints (inclusion and exclusion criteria) in the recruitment and participation of study participants. Therefore, the results need to be validated in a larger longitudinal, multinational group of patients with migraine. |
Regarding the lack of a control group, which was not stated in the manuscript in the first submission, we have added this limitation to the text. The text now reads as follows:
|
Fifth, as this was a longitudinal observational study on migraine symptoms, no control group was included within this study. |

Reviewer 4 Report
Comments and Suggestions for Authors
The manuscript entitled “A longitudinal comparative study of headache event symptomatology in patients with migraine between smartphone-based headache diary entries and clinician-led intake interviews” was interesting. The authors aimed to investigate similarity of headache event symptomatology and symptom agreement with clinician-led intake interviews of headache events registered via electronic headache diaries in patients with migraine. I think to improve the manuscript, further attentions in some parts are required.
1- The title is a bit long and needs to be shorter.
2- The aims of the study in the abstract and introduction section should be similar.
3- If the use of smart phone technology has been reported in other similar studies, the related literature should be reviewed and cited in the introduction section.
4- The rational for the current study should be explained in more details.
5- In the first paragraph of the methods section, please explain the research design and methodology.
6- Was the application available to the clinicians all the time? How was the dairy reviewed by the clinician?
7- What could be the difference between using and not using the smart phone? Do you think conducting an RCT is useful?
8- What are the implications of the research in practice?
9- Please follow the journal instructions for referencing style.
Author Response
Reviewer 4
The manuscript entitled “A longitudinal comparative study of headache event symptomatology in patients with migraine between smartphone-based headache diary entries and clinician-led intake interviews” was interesting. The authors aimed to investigate similarity of headache event symptomatology and symptom agreement with clinician-led intake interviews of headache events registered via electronic headache diaries in patients with migraine. I think to improve the manuscript, further attentions in some parts are required.
1- The title is a bit long and needs to be shorter.
We thank you for your valid comment. After debating the title with the co-authors, we have decided to submit a shorter and more concise title. The title now reads as follows:
|
Tracking Migraine Symptoms: A Longitudinal Comparison of Smartphone-Based Headache Diaries and Clinical Interviews |
2- The aims of the study in the abstract and introduction section should be similar.
We complied with your remark and have added the primary objective of the study from the introduction to the abstract.
|
Abstract: Background/Objectives: By leveraging the capabilities of a smartphone-based headache diary, the objective of the study is to determine the amount of agreement of migraine-associated symptomatology during headache events and the symptoms documented during clinician-led intake interviews. |
3- If the use of smart phone technology has been reported in other similar studies, the related literature should be reviewed and cited in the introduction section.
We have expanded the related literature in the introduction section. One study that was not mentioned before was the electronic headache diary study published by Verhagen et al. [1] where the authors compared differences in migraine attack characteristics between men and women longitudinally through electronic headache diaries and found that compared to attacks in men, both perimenstrual and non-perimenstrual migraine attacks are of longer duration and are more often accompanied by associated symptoms. Another study cited was the one by Vo et al.[2] who presented migraine symptomatology data from the smartphone application MigraineBuddy.
4- The rational for the current study should be explained in more details.
Thank you for this comment. In the introduction section we have tried to the best of our performance to explain to the reader why our current study is needed. First, we proclaim that there is a research gap on the knowledge of the heterogeneity and variability of intra-participant extended migraine symptomatology over time from longitudinal studies that apply smartphone-based headache diaries. Then, we explain that this study aims to fill this gap by leveraging the capabilities of a smartphone-based headache diary, to study the heterogeneity and variability of intraparticipant migraine symptomatology over time in patients with migraine. For good scientific rigor, we also propose a hypothesis for our study, namely that we expected moderate intake symptom agreement between clinician-led intake interviews and longitudinally registered headache events, with low to moderate intraparticipant event similarity between headache events, and a decline in intraparticipant similarity as the time between events increases.
5- In the first paragraph of the methods section, please explain the research design and methodology.
Thank you for this valuable remark. We have adapted the first paragraph of the methods section so that the first sentence reads as follows:
|
This was a 90-days observational, longitudinal, smartphone-based headache calendar study for participants diagnosed with migraine. |
6- Was the application available to the clinicians all the time? How was the dairy reviewed by the clinician?
Thank you for this question. The clinical information entered through the smartphone application was blinded for the clinician-investigators during the participation of the study subjects to avoid unnecessary information leakage. The only information available to the clinician-investigators at the time of study participation was the amount of compliance of the study subjects. If study subjects showed low compliance they would be contacted by the clinician-investigators to determine any problems or to encourage further participation. We added this information under subsection “2.2. Study Design” as follows:
|
During the participation of subjects in the study, the investigators had access to data on compliance with the smartphone application (to determine whether there were any problems or to encourage participants to continue with the trial), but did not have access to any clinically relevant data. |
7- What could be the difference between using and not using the smartphone? Do you think conducting an RCT is useful?
This study did not assess differences between using and not using the smartphone application, as we did not include a control group without the app. Conducting a randomized controlled trial (RCT) in future research could help establish the validity and potential advantages of smartphone-based headache diaries. However, at this early stage of our research, an RCT was not yet feasible. Additionally, comparing digital and non-digital headache diaries presents methodological challenges, particularly in ensuring standardization across formats.
8- What are the implications of the research in practice?
We believe commenting on the implications of research in clinical practice is very important and have included our analysis in the discussion section. Our analysis from this study brings forward interesting observations for the clinicians to understand the symptomatology and burden of patients with migraine. First, the research shows that intrapatient variability of headache symptomatology is present between different headache attacks. Second, the study underscores the importance of regularly evaluating and updating the list of symptoms for individual patients since newly discovered symptoms may come along during longitudinal analysis. For example, more than half of the patients (15/27, 55.6%) recognized at least one new ICHD-3 based symptom during the 90-days trial. Third, clinical interviews by physicians may assess the full spectrum of symptoms; however the agreement between recorded headache event symptomatology and clinician-led intake interview phenotyping is mild to moderate during real-world recordings.
9- Please follow the journal instructions for referencing style.
Thank you for this comment. We have checked all references and adapted them to the referencing style of the journal.
[1] Verhagen, Iris E., Britt W. H. Van Der Arend, Daphne S. Van Casteren, Saskia Le Cessie, Antoinette MaassenVanDenBrink, en Gisela M. Terwindt. ‘Sex Differences in Migraine Attack Characteristics: A Longitudinal E‐diary Study’. Headache: The Journal of Head and Face Pain 63, nr. 3 (maart 2023): 333-41. https://doi.org/10.1111/head.14488.
[2] Vo, Pamela, Nicolas Paris, Aikaterini Bilitou, Tomas Valena, Juanzhi Fang, Christel Naujoks, Ann Cameron, Frederic De Reydet De Vulpillieres, en Francois Cadiou. ‘Burden of Migraine in Europe Using Self-Reported Digital Diary Data from the Migraine Buddy© Application’. Neurology and Therapy 7, nr. 2 (december 2018): 321-32. https://doi.org/10.1007/s40120-018-0113-0.
Reviewer 5 Report
Comments and Suggestions for Authors
The manuscript presents valuable insights into the study of electronically registered headache events and their correlation with clinician-led interviews. The methodology and findings are clearly outlined, and the results contribute meaningfully to understanding migraine symptomatology. The methodology and discussion chapters are comprehensive and well-explained. Key points are clearly highlighted, including comparisons with previous studies, an in-depth discussion of the implications of the findings, and suggestions for future research.
However, the text would benefit from minor adjustments to punctuation and phrasing to improve clarity and readability. Additionally, the authors should revise the reference list and include more recent sources to accurately reflect the current state of development in electronic diaries. A few minor comments have been provided for improvement.
1. It is recommended to check the entire manuscript for grammar and punctuation. For example, in the abstract:
- Subject-verb agreement: In sentence 31, "study" is singular, so the verb should be "shows" instead of "show."
- Grammatical error with "90 days": It should be "a 90-day longitudinal study" instead of "90 days" because "90-day" is a compound adjective modifying "study." In academic writing, it’s often better to use a more direct and active construction, such as:
"We conducted a 90-day longitudinal study..."
"This study was a 90-day longitudinal investigation..."
2. The abstract is good as it effectively describes the methods and results of the study. However, it could be improved by explaining why the topic was chosen and highlighting its significance or potential impact.
3. Page 8 contains significant empty space. Consider revising the formatting to optimize the layout and ensure effective use of the page.
4. I recommend unifying the formatting of the tables to ensure consistency across the document. (Ex. Table 1 and 2)
5. The main text duplicates the information presented in Table 1. Please retain only the conclusions and key information about the participants, as the remaining details can be easily referenced in the table.
6. Since the use of electronic diaries is becoming an integral part of our reality at a rapid pace, it is recommended to use more recent references to support your viewpoint (from the past 5 years). This would strengthen the argument and reflect the latest trends and advancements in the field.
Author Response
Reviewer 5
The manuscript presents valuable insights into the study of electronically registered headache events and their correlation with clinician-led interviews. The methodology and findings are clearly outlined, and the results contribute meaningfully to understanding migraine symptomatology. The methodology and discussion chapters are comprehensive and well-explained. Key points are clearly highlighted, including comparisons with previous studies, an in-depth discussion of the implications of the findings, and suggestions for future research.
However, the text would benefit from minor adjustments to punctuation and phrasing to improve clarity and readability. Additionally, the authors should revise the reference list and include more recent sources to accurately reflect the current state of development in electronic diaries. A few minor comments have been provided for improvement.
- It is recommended to check the entire manuscript for grammar and punctuation. For example, in the abstract:
- Subject-verb agreement: In sentence 31, "study" is singular, so the verb should be "shows" instead of "show."
- Grammatical error with "90 days": It should be "a 90-day longitudinal study" instead of "90 days" because "90-day" is a compound adjective modifying "study." In academic writing, it’s often better to use a more direct and active construction, such as:
"We conducted a 90-day longitudinal study..."
"This study was a 90-day longitudinal investigation..."
Thank you for this kind remark. We acknowledge the grammatical errors and have corrected them where present.
- The abstract is good as it effectively describes the methods and results of the study. However, it could be improved by explaining why the topic was chosen and highlighting its significance or potential impact.
Thank you for this valid feedback and agree with your comment. We have adapted the background/objectives section of the abstract to highlight the following aspects. First, it mentions the potential impact of smartphone-based headache diaries to increase our knowledge on headache event symptomatology from patients with migraine. Second, it states the primary objective of the study is to determine the amount of agreement of migraine-associated symptomatology during headache events and the symptoms documented during clinician-led intake interviews.
- Page 8 contains significant empty space. Consider revising the formatting to optimize the layout and ensure effective use of the page.
Thank you for this remark. We recognized this issue and have deleted the excess whitespace.
- I recommend unifying the formatting of the tables to ensure consistency across the document. (Ex. Table 1 and 2)
We agree and have formatted all tables with the same layout to ensure the consistency.
- The main text duplicates the information presented in Table 1. Please retain only the conclusions and key information about the participants, as the remaining details can be easily referenced in the table.
We acknowledge this remark and have adapted the main text.
- Since the use of electronic diaries is becoming an integral part of our reality at a rapid pace, it is recommended to use more recent references to support your viewpoint (from the past 5 years). This would strengthen the argument and reflect the latest trends and advancements in the field.
The manuscript aims to both give a scope of new digital applications in headache research combined with the overall theme of digitization in medicine in general. We acknowledge that the manuscript at this point has not focused too strongly on the more general trend in medicine for the adoption of smartphone applications and electronic diaries. Therefore, we have searched and reviewed the literature on smartphone usage in medicine and electronic headache diaries in general. We have added the following references to the manuscript:
Daniëls, Naomi E M, Laura M J Hochstenbach, Catherine Van Zelst, Marloes A Van Bokhoven, Philippe A E G Delespaul, en Anna J H M Beurskens. ‘Factors That Influence the Use of Electronic Diaries in Health Care: Scoping Review’. JMIR mHealth and uHealth 9, nr. 6 (1 juni 2021): e19536. https://doi.org/10.2196/19536.
Iribarren, Sarah J, Tokunbo O Akande, Kendra J Kamp, Dwight Barry, Yazan G Kader, en Elizabeth Suelzer. ‘Effectiveness of Mobile Apps to Promote Health and Manage Disease: Systematic Review and Meta-Analysis of Randomized Controlled Trials’. JMIR mHealth and uHealth 9, nr. 1 (11 januari 2021): e21563. https://doi.org/10.2196/21563.
Sim, Ida. ‘Mobile Devices and Health’. New England Journal of Medicine 381, nr. 10 (5 september 2019): 956-68. https://doi.org/10.1056/NEJMra1806949.
These references have been cited after the following new sentence in the introduction:
|
Electronic diaries are being implemented widely in medicine [1-3]. In headache medicine, it is worth noting that certain studies have indicated a patient preference for electronic headache diaries over traditional paper-based methods …
[1] Daniëls, Naomi E M, Laura M J Hochstenbach, Catherine Van Zelst, Marloes A Van Bokhoven, Philippe A E G Delespaul, en Anna J H M Beurskens. ‘Factors That Influence the Use of Electronic Diaries in Health Care: Scoping Review’. JMIR mHealth and uHealth 9, nr. 6 (1 juni 2021): e19536. https://doi.org/10.2196/19536. [2] Iribarren, Sarah J, Tokunbo O Akande, Kendra J Kamp, Dwight Barry, Yazan G Kader, en Elizabeth Suelzer. ‘Effectiveness of Mobile Apps to Promote Health and Manage Disease: Systematic Review and Meta-Analysis of Randomized Controlled Trials’. JMIR mHealth and uHealth 9, nr. 1 (11 januari 2021): e21563. https://doi.org/10.2196/21563. [3] Sim, Ida. ‘Mobile Devices and Health’. New England Journal of Medicine 381, nr. 10 (5 september 2019): 956-68. https://doi.org/10.1056/NEJMra1806949.
Please note: reference numbers used in this box do not overlap with reference numbers used in the manuscript. |
Round 2
Reviewer 2 Report
Comments and Suggestions for Authors
Well done
Reviewer 3 Report
Comments and Suggestions for Authors
I thank authors for the work done
Reviewer 4 Report
Comments and Suggestions for Authors
I appreciate the authors for their time and efforts to revise the manuscript which has improved significantly.